# Conharmonic Curvature Inheritance in Spacetime of General Relativity †

**Musavvir Ali** [1,*]**, Mohammad Salman** [1] **and Mohd Bilal** [2]

[1]  Department of Mathematics, Aligarh Muslim University, Aligarh 202002, India; salman199114@gmail.com
[2]  Department of Mathematical Sciences, Faculty of Applied Sciences, Umm Al Qura University, Makkah P.O. Box 56199, Saudi Arabia; mohd7bilal@gmail.com
*  Correspondence: musavvirali.maths@amu.ac.in
†  2010 Mathematics Subject Classification: 53B20; 83C45; 53A45; 83C20.

**Abstract:** The motive of the current article is to study and characterize the geometrical and physical competency of the conharmonic curvature inheritance (Conh CI) symmetry in spacetime. We have established the condition for its relationship with both conformal motion and conharmonic motion in general and Einstein spacetime. From the investigation of the kinematical and dynamical properties of the conformal Killing vector (CKV) with the Conh CI vector admitted by spacetime, it is found that they are quite physically applicable in the theory of general relativity. We obtain results on the symmetry inheritance for physical quantities $(\mu, p, u_i, \sigma_{ij}, \eta, q_i)$ of the stress-energy tensor in imperfect fluid, perfect fluid and anisotropic fluid spacetimes. Finally, we prove that the conharmonic curvature tensor of a perfect fluid spacetime will be divergence-free when a Conh CI vector is also a CKV.

**Keywords:** curvature; symmetry; inheritance; Einstein spacetime





## 1. Introduction

Let $(V_4, g)$ be a spacetime, where $V_4$ is a four-dimensional connected smooth Hausdorff manifold and g is a smooth Lorentz metric of signature $(-, +, +, +)$. Let $\nabla$ be the Levi–Civita connection associated with g and $\mathcal{R}$ be the corresponding type (1, 3) Riemannian curvature tensor. The type (1, 3) Weyl conformal curvature tensor of $(V_4, g)$ is denoted by $\mathcal{C}$. The components of $\mathcal{R}$ and $\mathcal{C}$ are written as $R^h_{ijk}$ and $C^h_{ijk}$ and the Ricci tensor $R_{ij}$ and Ricci scalar $R$ are defined, in components, by $R_{ij} = R^h_{ihj}$ and $R = R_{ij}g^{ij}$, respectively.

In mathematics and theoretical physics, the study of spacetime symmetries is of great interest for contemporary researchers. In addition, the spacetime symmetries are very useful for finding the solutions to Einstein's field equation (EFE) if its existence occurs, and provide further intuition toward conservation laws of generators in dynamical systems [1]. Much interest has been shown in the various symmetries of the geometrical structures on $(V_4, g)$, and details are available in ([1–3]). Gravitational classification can be carried out through the help of geometrical symmetries of spacetime in general relativity. Moreover, motion/isometry or Killing symmetry is one of the most primary symmetries of a spacetime. This is defined along a vector field under the condition that the Lie derivative of metric tensor vanishes.

An elegant restructuring form of classical mechanics is finalized by the general theory of relativity. This theory wraps the time and the space co-ordinates into a single continuum, called as spacetime. This theory is also called as the theory of gravitation in spacetime, which is described by the Einstein's field equation and these equations describe a system of ten coupled highly nonlinear PDEs, given as the following

$$R_{ij} - \frac{1}{2}Rg_{ij} = \kappa T_{ij}, \tag{1}$$

where $T_{ij}$ denotes the components of the stress-energy tensor and $\kappa$ is the gravitational constant. In the field Equation (1), the left part depicts the geometrical meaning of spacetime, whereas the right part describes the physical significance of the spacetime of general relativity.

The study of spacetime symmetries is an important tool in finding the exact solution of the system (1). The spacetime symmetries play a pivotal role in understanding the relationship between matter and geometry by EFE. The different classes of spacetime symmetries, such as the isometries, homothetic motion, conformal motion, curvature symmetry, curvature inheritance symmetry, Ricci symmetry, Ricci inheritance symmetry, matter collineations, matter inheritance collineations, conharmonic symmetries, semi-conformal symmetry, etc., are well known in the literature ([1–6]). The spacetime symmetries are important not only in finding the exact solutions of EFE, but also in providing spacetime classifications along with an invariant basis (preferably, the basis of null tetrad can be chosen). The spacetime symmetries are also a popular tool in investigating many conservation laws in the theory of general relativity [5]. Moreover, certain geometrical and physical notions are also described by spacetime symmetries, such as the conservation of linear momentum, angular momentum and energy [7]. The symmetries regarding spacetime $(V_4, g)$ are determine by the following mathematical equation [8]:

$$\pounds_\xi \Omega = 2\alpha \Omega, \tag{2}$$

where $\pounds_\xi$ stands for the Lie derivative operator, with respect to the vector field $\xi^i$, $\alpha$ is some smooth scalar function on the spacetime and $\Omega$ is any of the physical quantities $(\mu, p, u_i, \sigma_{ij}, \eta, q_i)$, where $\mu, p, u_i, \sigma_{ij}, \eta, q_i$ are the energy density, the isotropic pressure, the velocity vector, the shear tensor, the shear viscosity coefficient and the energy flux vector, respectively, and geometrical quantities, such as the components of the metric tensor $(g_{ij})$, Riemannian curvature tensor $(R^h_{ijk})$, Ricci tensor $(R_{ij})$, conharmonic curvature tensor $(Z^h_{ijk})$, contracted conharmonic curvature tensor $(Z_{ij})$, energy momentum tensor $(T_{ij})$, etc. The most primary symmetry on $(V_4, g)$ is motion (M), which is obtained by setting $\Omega = g_{ij}$ and $\alpha = 0$ in Equation (2). Then, Equation (2) will be called the Killing equation, and the vector satisfying it is known as the Killing vector. Equation (2) can also be explicitly written as the following:

$$\xi^k g_{ij,k} + g_{ik}\xi^k_{,j} + g_{jk}\xi^k_{,i} = 0, \tag{3}$$

where the subscript comma $(,)$ stands for the partial differentiation, with respect to the coordinates $(x^i)$ in the spacetime.

The gravitational field consists of two parts viz., the free gravitational part and the matter part, which is described by the Riemannian curvature tensor in the general theory of relativity. The connection between these two parts is explained through Bianchi's identities [9]. The principal aim of all investigations in gravitational physics is focused on constructing the gravitational potential (metric) satisfying the Einstein field equations.

In the present research paper, we raise the following fundamental problem: how are the geometrical symmetries of the spacetime $(V_4, g)$ associated with the conharmonic curvature symmetry vector field, under the condition that this vector is inherited by some of the source terms of the energy-momentum tensor in the field equations? In this paper, we discuss the conharmonic curvature inheritance symmetry with respect to conformal motion, conharmonic motion and source terms of perfect, imperfect and anisotropic fluid spacetime. Our present work is mainly influenced by the work carried out towards the symmetries, such as the curvature inheritance, Ricci inheritance, and matter inheritance on the semi-Riemannian manifold. This concept of symmetry inheritance was initiated in 1989 by Coley and Tupper [10] for the special conformal Killing vector (SCKV), and was then further studied in 1990 for CKV ([11,12]). In 1992 and 1993, K. L. Duggal introduced the concept of inheritance symmetry for the curvature tensor of Riemannian spaces with physical applications to the fluid spacetime of general relativity ([2,13]).



The above abundant work motivated us to inquire about the inheritance symmetry of the conharmonic curvature tensor in spacetime. The conharmonic curvature inheritance symmetry is defined through Equation (2), where $\Omega$ is replaced by the conharmonic curvature tensor. The structure of our manuscript is as follows: the preliminaries are given in Section 2. In Section 3, we elaborate on the concept of curvature inheritance symmetry with some of the related results. In Section 4, we derive the relationship of symmetry inheritance with other known symmetries, such as both conformal motion and conharmonic motion in general and Einstein spacetime. We have established some important results as a witness to the physical application of the Conh CI symmetry in spacetime for perfect, imperfect and anisotropic fluid in Section 5. Finally, Section 6 is a brief conclusion. Furthermore, in an attempt to support our study, which is related to the solution of EFE and conservation law of generators, we have constructed some non-trivial examples that are embedded in the Appendix A after the conclusion.

## 2. Preliminaries

If the Lie derivative of the Riemann curvature tensor, along a vector field $\xi$, vanishes i.e., $\pounds_{\xi} R^h_{ijk} = 0$, then it is called a curvature collineation (CC), which was introduced by Katzin et al. [5] in 1969. The Ricci collineation (RC) is obtained by the contraction of the expression $\pounds_{\xi} R^h_{ijk} = 0$ and is given by $\pounds_{\xi} R_{ij} = 0$.

Conformal motion (Conf M) along a vector $\xi$ is defined in the following manner:

$$\hbar_{ij} = \pounds_{\xi} g_{ij} = 2\alpha g_{ij}, \qquad \alpha = \alpha(x^i), \tag{4}$$

where $\alpha$ is the conformal function on $(V_4, g)$ and $\xi$ is called the conformal Killing vector (CKV). If $\alpha$ satisfies the condition

$$\alpha_{;ij} = 0 \quad and \quad \alpha_{;i} \neq 0, \tag{5}$$

then $\xi$ is the special conformal Kiling vector field (SCKV), where the semi-colon (;) represents the covariant differentiation. The next subclass is homothetic motion (HM), if $\alpha_{;i} = 0$ and motion (M), if $\alpha = 0$.

The projective collineation (PC) satisfies $\pounds_{\xi} W^h_{ijk} = 0$, where $W^h_{ijk}$ denotes the Weyl projective curvature tensor in $(V_4, g)$ and is defined as follows:

$$W^h_{ijk} = R^h_{ijk} + \frac{1}{3} [\delta^h_j R_{ik} - \delta^h_k R_{ij}]. \tag{6}$$

The projective collineation is defined in another way by a vector field $\xi$ satisfying

$$\pounds_{\xi} \Gamma^i_{jk} = \delta^i_j \rho_k + \delta^i_k \rho_j, \tag{7}$$

where $\rho_i = \partial_i \rho$ for a scalar field $\rho$, $\Gamma^i_{jk}$ are the components of the Christoffel symbol of the Riemannian metric g and $\delta^i_j$ stands for the Kronecker delta.

The curvature inheritance (CI) ([2,3]) along a vector field $\xi$ is defined on the Riemannian space as:

$$\pounds_{\xi} R^h_{ijk} = 2\alpha R^h_{ijk}, \tag{8}$$

where $\alpha$ is an inheritance function of spacetime coordinates and vector field $\xi$ is called the curvature inheritance vector and is abbreviated as (CIV). Similarly, the Ricci inheritance (RI) is defined as

$$\pounds_{\xi} R_{ij} = 2\alpha R_{ij}, \tag{9}$$

The vector field $\xi$ is called the Ricci inheritance vector (RIV). As we know that every CIV is a RIV, and from [2], we have

$$\pounds_{\xi} R^i_j = 2\alpha R^i_j - R^i_l \hbar^l_j \tag{10}$$

and

$$\pounds_\xi R = 2\alpha R - R_{ij}\hbar^{ij}, \tag{11}$$

where

$$\hbar_{ij} = \pounds_\xi g_{ij} = \xi_{i;j} + \xi_{j;i}. \tag{12}$$

The study of the exact solutions of the Einstein field equation and related conservation laws is carried out with symmetry assumptions on spacetime. In addition, such a study is carried out by numerous authors by adopting various methods (cf., [1,5,14]).

The introduction of the conharmonic transformation as a subgroup of the conformal transformation was given by Ishii [15] and defined the following transformation,

$$\overline{g_{ij}} = g_{ij}e^{2\sigma}, \tag{13}$$

where $\sigma$ stands for the scalar function and also the following condition holds:

$$\sigma^i_{;i} + \sigma_{;i}\sigma^i = 0. \tag{14}$$

On spacetime $(V_4, g)$, a quadratic Killing tensor is a generalization of a Killing vector and is defined as a second-order symmetric tensor $A_{ij}$ [16] satisfying the condition:

$$A_{ij;k} + A_{jk;i} + A_{ki;j} = 0. \tag{15}$$

A vector field $\xi$ in a semi-Riemannian space is said to generate a one-parameter group of curvature collineations [17] if it satisfies:

$$\pounds_\xi R = 0. \tag{16}$$

A Riemannian space is conformally flat [18] if

$$C^h_{ijk} = 0, \quad (n > 3). \tag{17}$$

A Riemannian space is conharmonically flat [16] if

$$Z^h_{ijk} = 0, \quad (n > 3). \tag{18}$$

### 3. Conharmonic Curvature Inheritance Symmetry

A $(1, 3)$-type conharmonic curvature tensor $Z^h_{ijk}$, which is unaltered under the conharmonic transformation (13) and (14), can be explicitly expressed as the following equation [19]:

$$Z^h_{ijk} = R^h_{ijk} + \frac{1}{2}(\delta^h_j R_{ik} - \delta^h_k R_{ij} + g_{ik}R^h_j - g_{ij}R^h_k). \tag{19}$$

We introduce the notion of conharmonic curvature inheritance symmetry as follows.

**Definition 1.** *On spacetime $V_4$ with Lorentzian metric g, a smooth vector field $\xi$ is said to generate a conharmonic curvature inheritance symmetry if it satisfies the following equation:*

$$\pounds_\xi Z^h_{ijk} = 2\alpha Z^h_{ijk}, \tag{20}$$

*where $\alpha = \alpha(x^i)$ is an inheritance function.*

**Proposition 1.** *If a spacetime $(V_4, g)$ admits the following symmetry inheritance equations:*

*(a)* $\pounds_\xi R^h_{ijk} = 2\alpha R^h_{ijk}$,
*(b)* $\pounds_\xi g_{ij} = 2\alpha g_{ij}$,

*then that spacetime necessarily admits Conh CI along a vector field $\xi$.*

**Proof.** The proof is obtained directly by taking the Lie derivative of the Equation (19), and using above symmetry inheritance equations we have $\pounds_\xi Z^h_{ijk} = 2\alpha Z^h_{ijk}$. Thus, spacetime admits Conh CI along a vector field $\xi$.   □

**Example 1.** *Consider the following line element of a de Sitter spacetime:*

$$ds^2 = -dt^2 + e^{2\lambda t}(dx^2 + dy^2 + dz^2), \tag{21}$$

*where $\lambda$ is a constant. This line element admits a proper CKV, $\xi^i = (e^{\lambda t}, 0, 0, 0)$, for which $\alpha = \lambda e^{\lambda t}$. A straightforward computation of the components $R^h_{ijk}$, and then taking the Lie derivative with respect to $\xi$, indicates that $\xi$ is a CIV and, therefore, an RIV. Thus, this example of $(V_4, g)$ with the above metric is compatible with Proposition 1, i.e., de Sitter spacetime satisfying the Conh CI symmetry.*

In this research article, we are considering the inheritance function as being the same as the conformal function. If $\alpha = 0$, then (20) reduces to $\pounds_\xi Z^h_{ijk} = 0$, which is called conharmonic curvature collineation (Conh CC) [4]. Contracting (20), we obtain

$$\pounds_\xi Z_{ij} = 2\alpha Z_{ij}, \tag{22}$$

where $Z_{ij}$ denotes the contracted conharmonic curvature tensor on a spacetime $(V_4, g)$ [16], and it is invariant under the transformation (13).

**Definition 2.** *On spacetime $(V_4, g)$, a smooth vector field $\xi$ is said to generate a contracted conharmonic curvature inheritance symmetry if it satisfies the Equation (22).*

Thus, in general, every Conh CI vector is a contracted Conh CI vector, but its converse may not hold. In particular, if $\alpha = 0$, (22) reduces to

$$\pounds_\xi Z_{ij} = 0. \tag{23}$$

**Definition 3.** *A vector field $\xi$ satisfying (23) is called a contracted conharmonic curvature collineation vector field.*

If $\alpha \neq 0$, then a vector field $\xi$ satisfying (22) is called a proper contracted Conh CI vector. Contracting Equation (19), we obtain

$$Z_{ij} = -\frac{1}{2} g_{ij} R. \tag{24}$$

**Lemma 1.** *If a spacetime $(V_4, g)$ admits the contracted conharmonic curvature tensor, then the scalar curvature of the spacetime $(V_4, g)$ will be constant.*

**Proof.** Recently, ref. [16] U. C. De, L. Velimirovic and S. Mallick studied the characteristics of the contracted conharmonic curvature tensor ($Z_{ij}$) as follows: "In a spacetime, the contracted conharmonic curvature tensor is a quadratic Killing tensor", or it can be written as $Z_{ij;k} + Z_{jk;i} + Z_{ki;j} = 0$ with the use of Equation (15). They also stated that "a necessary and sufficient condition for contracted conharmonic curvature tensor [to] be a quadratic Killing tensor is that the scalar curvature of the spacetime be constant". Now, using Equation (24) in $Z_{ij;k} + Z_{jk;i} + Z_{ki;j} = 0$, we obtain

$$R = \text{constant}. \tag{25}$$

This completes the proof.   □

**Remark 1.** *On the Lie derivative of* (24) *along a proper conformal Killing vector field $\xi$* (4), *and using* (25), *we can easily show that Equation* (22) *is well defined on spacetime* $(V_4, g)$.

**Theorem 1.** *If a spacetime* $(V_4, g)$ *admits Conh CI along a vector field $\xi$, then the following identities hold:*

(a) $\pounds_\xi Z_{ij} = 2\alpha Z_{ij}$,
(b) $\pounds_\xi Z^i_j = 2\alpha Z^i_j - Z^k_j \hbar^i_k$,
(c) $\pounds_\xi R = 0$.

**Proof.** Contracting Equation (20), we obtain $\pounds_\xi Z_{ij} = 2\alpha Z_{ij}$, which proves (a) and implies that every Conh CI is a contracted Conh CI. The proof of (b) follows by $\pounds_\xi Z_{ij} = \pounds_\xi(g_{jk} Z^k_i)$ and the use of Equation (4), which leads to $\pounds_\xi Z_{ij} = g_{jl}(\pounds_\xi Z^l_i + \hbar^l_k Z^k_i)$. Now, comparing with part (a) and the rearrangement, we obtain the required result (b). Since spacetime $(V_4, g)$ also admits the conharmonic curvature tensor, and, in general, every conharmonic curvature tensor$(Z^h_{ijk})$ is a contracted conharmonic curvature tensor $(Z_{ij})$., under the hypothesis of Lemma 1, this implies that the scalar curvature is constant. Now, following the Lie derivative of Equation (25) proves part (c). $\square$

**Remark 2.** *Clearly, under the hypothesis of Theorem* 1, *spacetime* $(V_4, g)$ *generates a one-parameter group of curvature collineation* [17].

In the empty spacetime $(R_{ij} = 0)$, the tensors $R^h_{ijk}$ and $Z^h_{ijk}$ are identical. This implies that, in empty spacetime, Conh CI reduces to curvature inheritance symmetry.

Now, here, we obtain the result on the symmetry inheritance for the spacetime admitting the conformal curvature tensor under consideration of Conh CI.

**Theorem 2.** *If a spacetime* $(V_4, g)$ *admits the conharmonic curvature inheritance symmetry along a vector field $\xi$, the conformal curvature tensor satisfies the symmetry inheritance property.*

**Proof.** The conformal curvature tensor is

$$C^h_{ijk} = R^h_{ijk} + \frac{1}{2}(\delta^h_j R_{ik} - \delta^h_k R_{ij} + R^h_j g_{ik} - R^h_k g_{ij}) + \frac{R}{6}(g_{ij}\delta^h_k - g_{ik}\delta^h_j), \quad (26)$$

and this expression is also written in terms of $Z^h_{ijk}$ and $Z_{ij}$ as

$$C^h_{ijk} = Z^h_{ijk} + \frac{1}{3}(Z_{ik}\delta^h_j - Z_{ij}\delta^h_k). \quad (27)$$

Taking the Lie derivative of (27) and using (20) and (22), we obtain

$$\pounds_\xi C^h_{ijk} = 2\alpha C^h_{ijk}. \quad (28)$$

This completes the proof. $\square$

Now, we state Theorem 3(e) from [2], i.e., "If a spacetime $(V_4, g)$ admits a CI, then the following identity holds:

$$\pounds_\xi C^h_{ijk} = 2\alpha C^h_{ijk} + D^h_{ijk}, \quad (29)$$

where

$D^h_{ijk} = \frac{1}{2}[R^h_j \hbar_{ik} - R^h_k \hbar_{ij} + R^l_k \hbar^h_l g_{ij} - R^l_j \hbar^h_l g_{ik}] + \frac{1}{6}[\delta^h_k(R\hbar_{ij} - R'g_{ij}) - \delta^h_j(R\hbar_{ik} - R'g_{ik})]$

and $R' = 2R^i_j \xi^j_{;i}$."

The above result raises the following open problem [13]: "Find condition on $(V_4, g)$, with a proper CI symmetry such that $D^h_{ijk}$ vanishes". From Theorem 2, we solve the above open problem for the spacetime $(V_4, g)$ to admit proper Conh CI. If a spacetime admits Conh CI symmetry, then (29) is singled out as free from the term $D^h_{ijk}$. At this point, we

mention that Conh CI is very important in the comparison of the CI symmetry; it restricts $(V_4, g)$ to a very limited geometrical use, as well as physical use.

**Remark 3.** *The Theorem 2 gives us a motivation of the Conh CI symmetry of spacetime, since it implies the conformal curvature inheritance symmetry (28). On the other hand, the CI does not imply the conformal curvature inheritance symmetry.*

Now, we shall investigate the role of such a symmetry inheritance for the spacetime admitting the Weyl projective curvature tensor $(W_{ijk}^h)$.

**Theorem 3.** *Under the hypothesis of Proposition 1, if a spacetime $(V_4, g)$ admits the Weyl projective tensor with Conh CI along a vector field $\xi$, then the Weyl projective tensor also holds the symmetry inheritance property.*

**Proof.** Let a spacetime $(V_4, g)$ admit the Weyl projective tensor with a Conh CI along a vector field $\xi$; this tensor is expressed as

$$W_{ijk}^h = R_{ijk}^h + \frac{1}{3}[\delta_j^h R_{ik} - \delta_k^h R_{ij}]. \tag{30}$$

Taking the Lie derivative of (30), we have

$$\pounds_\xi W_{ijk}^h = \pounds_\xi R_{ijk}^h + \frac{1}{3}[\delta_j^h(\pounds_\xi R_{ik}) - \delta_k^h(\pounds_\xi R_{ik})]. \tag{31}$$

Further, from Proposition 1, $(V_4, g)$ also admits a CIV and RIV, so we have

$$\pounds_\xi W_{ijk}^h = 2\alpha W_{ijk}^h. \tag{32}$$

This completes the proof. □

**Theorem 4.** *If a spacetime admits a Conh CI along vector $\xi$, then it satisfies the condition*

$$\hbar_{ij;kl} - \hbar_{ij;lk} = 0. \tag{33}$$

**Proof.** As we know that the conharmonic curvature tensor satisfies the identity

$$Z_{jklm} + Z_{kjlm} = 0. \tag{34}$$

we can also write

$$Z_{klm}^i g_{ij} + Z_{jlm}^i g_{ik} = 0. \tag{35}$$

Taking the Lie derivative of (35), using Equations (20) and (4), we obtain

$$Z_{klm}^i \hbar_{ij} + Z_{jlm}^i \hbar_{ik} = 0. \tag{36}$$

Now, using the expression of $Z_{ijk}^h$ and Equation (4) in Equation (36), we obtain

$$R_{klm}^i \hbar_{ij} + R_{jlm}^i \hbar_{ik} = 0. \tag{37}$$

Applying the Ricci identity [9] on (37), we obtain (33), which completes the proof. □

**Remark 4.** *If we multiply by $\sqrt{g} g^{il} g^{jk}$ in (33), we obtain the Komar's identity [20]*

$$[\sqrt{g}(\xi^{i;j} - \xi^{j;i})]_{;ji} = 0 = [[\sqrt{g}(\xi^{i;j} - \xi^{j;i})]_{;j}]_{;i}, \tag{38}$$

*where g= $det(g_{ij})$ and Equation (33) is a necessary condition for a Conh CI and is also independent of the inheritance function $\alpha$ of (20), and is the same as for CC and CI.*

*Komar's identity directly interplays in the conservation law generator in general relativity [20], where $(V_4, g)$ admits curvature symmetry properties. As Komar's identity holds for all vector fields $\xi$ on $V_4$*

$$\hbar_{ij;kl} - \hbar_{ij;lk} = 0, \tag{39}$$

*for a CC, CI plays no restriction on this symmetry vector $\xi$. Hence, Conh CI are the necessary symmetry properties of spacetime $(V_4, g)$ that are embraced by the group of general curvilinear co-ordinate transformations in $V_4$.*

Furthermore, following the condition that Equation (39) is independent of the scalar function $\alpha$ in a (20), we observe that Conh CI retains this conharmonic transform characteristics of the Conh CC of the spacetime geometry.

## 4. Relationship of Conh CI with Other Symmetries of Spacetime

In this section, we describe a relationship of Conh CI with other well-known symmetries of spacetime, such as conformal motion (Conf M) and conharmonic motion (Conh M). We also obtain many results on the relationship between these symmetries. First, we give the introduction and its characteristics' results of those symmetries of spacetime, which are required for the development of the present research work.

### 4.1. Conformal Motion

A spacetime $(V_4, g)$ admits Conf M [5] along a (CKV) $\xi$ if the following equation is satisfied,

$$\hbar_{ij} = \pounds_\xi g_{ij} = 2\alpha g_{ij}, \tag{40}$$

where $\alpha = \frac{1}{4}\xi^k_{;k}$. If $\alpha_{;ij} = 0, \alpha_{;i} \neq 0$, then $\xi$ is called a (SCKV). Other CKVs are the homothetic motion (HM) if $\alpha_{;i} = 0, \alpha \neq 0$ and the motion (M) if $\alpha = 0$.

A $(V_4, g)$ is said to admit a conformal collineation (Conf C) if a vector exists $\xi$ such that

$$\pounds_\xi \Gamma^i_{jk} = \delta^i_j \alpha_{;k} + \delta^i_k \alpha_{;j} - g_{jk} g^{il} \alpha_{;l}, \tag{41}$$

and along the vector field $\xi$, a Weyl conformal collineation (W Conf C) is said to be admitted by a spacetime if

$$\pounds_\xi C^h_{ijk} = 0. \tag{42}$$

Every Conf M implies Conf C and W Conf C, but the converse is not necessarily true. Further, we have the condition [2],

$$\pounds_\xi R_{ij} = -\Box \alpha g_{ij} - 2\alpha_{;ij}, \tag{43}$$

where $\Box$ is the Laplacian–Beltrami operator defined by $\Box \alpha = g^{ij}\alpha_{;ij}$.

**Theorem 5.** *If a Conh CI vector $\xi$ is also a conformal Killing vector (CKV) on a spacetime $(V_4, g)$, then*

$$(a)\ \alpha_{;ij} = -\frac{1}{3}\alpha R_{ij},$$

$$(b)\ \Box \alpha + \frac{1}{3}\alpha R = 0, \tag{44}$$

$$(c)\ \alpha_{;ij} = -\frac{1}{3}\alpha(3R_{ij} + Z_{ij}).$$

**Proof.** For a Conh CI, Proposition 1 implies the following equation:

$$\pounds_\xi R_{ij} = 2\alpha R_{ij}. \tag{45}$$

Since $\xi$ is also a CKV, it must satisfy (43). Thus, comparing Equations (43) and (45), we obtain

$$2\alpha R_{ij} = -\Box \alpha g_{ij} - 2\alpha_{;ij}, \tag{46}$$

after simplification, (46) reduces to the (44)(a). Multiplying both side of (44)(a) by $g^{ij}$, we obtain (44)(b). The proof of (44)(c) follows from Equations (46), (44)(b) and (24). □

**Remark 5.** *All of the results of the Theorem 5 are very useful in the further study of conharmonic motion (Conh M) and Conh CI symmetry in the context of the space time of general relativity. They have a direct role as applications in the anisotropic, perfect and imperfect fluid spacetimes.*

**Corollary 1.** *If a spacetime $(V_4, g)$ admits the Conh CI with $\xi$ as a conformal Killing vector, then the conformal curvature tensor vanishes.*

**Proof.** The proof follows from Equations (28), (42) and (17). It may be noted that, for a CKV, the conformal curvature tensor vanishes in $(V_4, g)$, i.e., spacetime is conformally flat. □

**Example 2.** *It is well known that the Weyl conformal curvature tensor $C^h_{ijk} = 0$ if the spacetime is conformally flat. By definition, the line element of a conformally flat spacetime can be written as*

$$ds^2 = f^2(t, x, y, z)(-dt^2 + dx^2 + dy^2 + dz^2).$$

*All conformally flat solutions with a perfect fluid, an electromagnetic field or a pure radiation field are known.*

**Corollary 2.** *If spacetime admits M, HM or SCKV, then Conh CI must be a Conh CC.*

**Proof.** In particular, a relationship of Conh CI with curvature collineation (CC) is described by the hypothesis of Theorem 5. Since, in a $(V_4, g)$, every motion (M) is a CC; therefore, every HM and SCKV is also a CC. Thus, Conh CI must be a Conh CC when taking the Lie derivative of Equation (19). □

Now, we discuss Conh CI in Einstein spaces.

**Theorem 6.** *Every proper Conh CI in an Einstein space with a non-zero scalar curvature is a proper Ricci inheritance.*

**Proof.** Let $(V_4, g)$ be an Einstein spacetime with a non-zero scalar curvature,

$$R_{ij} = \frac{R}{4} g_{ij}, \qquad R = constant. \tag{47}$$

Comparing Equation (47) with (24), we obtain

$$Z_{ij} = -2R_{ij}. \tag{48}$$

Taking the Lie derivative of (48) and using (22), we obtain

$$\pounds_\xi R_{ij} = 2\alpha R_{ij}. \tag{49}$$

Thus, the Einstein spaces admit the Ricci inheritance symmetry. □

**Corollary 3.** *Under the hypothesis of Theorem 6, if $\xi$ is a Ricci inheritance vector (RIV), then the associated Conh CI must be a proper Conf M with the conformal function $\alpha$.*

**Theorem 7.** *An Einstein spacetime admits Conh CI along a vector field $\xi$ if $\xi$ is a curvature inheritance vector (CIV).*

**Proof.** Let $(V_4, g)$ be an Einstein spacetime that admits a Conh CI vector $\xi$, i.e.,

$$\pounds_\xi Z^h_{ijk} = 2\alpha Z^h_{ijk}. \tag{50}$$

Now, using Theorem 6, we obtain

$$\pounds_\xi R_{ij} = 2\alpha R_{ij}. \tag{51}$$

Again, by virtue of Corollary 3,

$$\pounds_\xi g_{ij} = 2\alpha g_{ij}. \tag{52}$$

Now, $R^h_{ijk}$ can be expressed as follows:

$$R^h_{ijk} = Z^h_{ijk} - \frac{1}{2}(\delta^h_j R_{ik} - \delta^h_k R_{ij} + R^h_j g_{ik} - R^h_k g_{ij}). \tag{53}$$

Taking the Lie derivative of (53) and using Equations (50)–(52), we obtain

$$\pounds_\xi R^h_{ijk} = 2\alpha R^h_{ijk}. \tag{54}$$

Thus, Conh CI reduces to a CI.

The converse part is also obvious: if an Einstein spacetime admits CI symmetry along a vector field $\xi$, i.e.,

$$\pounds_\xi R_{ij} = 2\alpha R_{ij}. \tag{55}$$

From Duggal (cf., [2], p. 2992), if an Einstein spacetime admits proper CI, then the spacetime also admits proper Conf M, i.e.,

$$\pounds_\xi g_{ij} = 2\alpha g_{ij}. \tag{56}$$

taking the Lie derivative of Equation (19) and using Equations (8)–(10) and (56). Thus, we conclude that CI reduces to a Conh CI. $\square$

Now, we derive a necessary condition for Conh CI symmetry, admitted by a spacetime that is not an Einstein spacetime.

**Theorem 8.** *A necessary condition for spacetime $(V_4, g)$ admitting Conh CI is that the spacetime admits both CI and Conf M together.*

**Proof.** We let spacetime $(V_4, g)$ admit $\xi$ as the CIV on it; this implies that $\xi$ satisfies Equations (8) and (9). Moreover, it is known (cf., [2], p. 2991) that if the spacetime admits CI, then the following identity holds:

$$\pounds_\xi R^i_j = 2\alpha R^i_j - R^i_l \hbar^l_j. \tag{57}$$

Taking the Lie derivative of (19) and using Equations (8), (9), (57) and the equation of Conf M, we see that the spacetime admits Conh CI symmetry along the vector field $\xi$, i.e.,

$$\pounds_\xi Z^h_{ijk} = 2\alpha Z^h_{ijk}. \tag{58}$$

This completes the proof. $\square$

**Remark 6.** *In general, the converse of Theorem 8 is not true, while the converse holds if $(V_4, g)$ is an Einstein spacetime. We conclude that the advantage of the selection of the Einstein spacetime is the relaxation of the condition for Conf M.*

**Example 3.** *Let $(V_4, g)$ be an Einstein spacetime admitting a Conh CI, which implies admitting the CI, as well as Conf M. Then, following on from Corollary 3 using (47) in the first result of Theorem 5, we obtain*

$$\alpha_{;ij} = (-\frac{\alpha R}{12})g_{ij}, \tag{59}$$

*where $\alpha$ and $R$ are both scalar functions of spacetime co-ordinates. We consider the single scalar function $\phi$ instead of $(-\frac{\alpha R}{12})$ in (59), and then we obtain*

$$\alpha_{;ij} = \phi g_{ij}. \tag{60}$$

*Petrov [21] referred to a finding of Sinyukov [22] that explains that, if a spacetime $(V_4, g)$ admits a vector field $\phi_{;i}$ satisfying (60) for $\phi \neq 0$, then a system of co-ordinates exists where the metric has the form:*

$$ds^{*2} = g_{11}dx^1 dx^1 + (\frac{1}{g_{11}})\Gamma_{ab}(x^2, x^3, x^4)dx^a dx^b, \tag{61}$$

*where $a, b \neq 1$ and $g_{11} = [2\int \phi(x^1)dx^1 + C]^{-1}$, and the arbitrary function $\phi = \phi(x^1)$.*

The above example of $(V_4, g)$ with metric (61) is well suited for Theorems 6–8 and Corollary 3.

*4.2. Conharmonic Motion*

Abdussatar and Babita Dwivedi [4] introduced a new type of conformal symmetry called conharmonic symmetry. Conharmonic motion (Conh M) is defined through a definition of Conf M (40) as follows:

$$\Box \alpha = g^{ij}\alpha_{;ij} = 0, \qquad \alpha_{;ij} \neq 0. \tag{62}$$

Similarly, a Conh CC is defined through Conf C if Equation (41) holds with the condition (62). If a vector field $\xi$ satisfies

$$£_\xi Z^h_{ijk} = 0, \tag{63}$$

then $(V_4, g)$ admits a conharmonic curvature collineation (Conh CC), and $\xi$ is also known as a conharmonic Killing vector (Conh KV). Clearly, every conharmonic motion is a Conh CC, but the converse is not true in general. From Equation (28), it is evident that every Conh CC is a W Conf C. Every Conh M satisfies

$$£_\xi R^h_{ijk} = \delta^h_j \alpha_{;ik} - \delta^h_k \alpha_{;ij} + \alpha^h_{;j}g_{ik} - \alpha^h_{;k}g_{ij}, \tag{64}$$

$$£_\xi R_{ij} = -2\alpha_{;ij}, \tag{65}$$

$$£_\xi R^j_k = -2\alpha^j_{;k} - 2\alpha R^j_k, \tag{66}$$

$$£_\xi R = -2\alpha R. \tag{67}$$

Multiplying by $g^{ij}$ in Equation (65) and in view of (62), we observe that

$$g^{ij}£_\xi R_{ij} = 0. \tag{68}$$

Thus, we can say that every Conh M reduces to a contracted Ricci collineation, but that the converse is not true.

We also have the following:

**Theorem 9.** *If the spacetime $(V_4, g)$ admits a Conh CI as well as a proper Conh M, then the scalar curvature of spacetime vanishes.*

**Proof.** Comparing (67) with Equation (c) in Theorem 1, we obtain $-2\alpha R = 0$; that is, the scalar curvature $R$ of the spacetime vanishes. $\square$

**Remark 7.** *Now, here, we will be discussing the motivation of Theorem 9. From [2], every CIV is a RIV , but the converse is not true in general (for further details, see Theorem 3.2 in [4] and Propositions 1 and 2 in [23]). If spacetime admits a Conh CIV that is also a RIV, then every RIV is a CIV. This information was not available to Sharma, R. and Duggal, K.L. et al. [23] in 1994, when they introduced CI. This is certainly an improvement over the use of Conh curvature symmetries because the proper CIV exists together with the proper CKV, which has greater physical significance.*

Moreover, we have the following result:

**Theorem 10.** *If $\xi$ is a Conh CIV as well as a RIV, then*

$$\pounds_{\xi} R^h_{ijk} = 2\alpha R^h_{ijk}. \tag{69}$$

**Proof.** Using the Lie derivation of Equation (19) with respect to $\xi$, and then using the inherited symmetry properties of $R_{ij}$, $Z^h_{ijk}$ and $g_{ij}$, we obtain

$$\pounds_{\xi} R^h_{ijk} = 2\alpha R^h_{ijk}, \tag{70}$$

i.e., the Riemann curvature tensor is inherited in spacetime. $\square$

Next, we also have

**Theorem 11.** *If a spacetime $(V_4, g)$ admits proper Conh CI along a conharmonic Killing vector $\xi$, then that spacetime is conharmonically flat.*

**Proof.** $Z^h_{ijk}$ is expressed as

$$R^h_{ijk} + \frac{1}{2}(\delta^h_j R_{ik} - \delta^h_k R_{ij} + g_{ik} R^h_j - g_{ij} R^h_k). \tag{71}$$

Taking the Lie derivative of (71), along the vector field $\xi$,

$$\pounds_{\xi} Z^h_{ijk} = \pounds_{\xi}(R^h_{ijk}) + \frac{1}{2}(\delta^h_j \pounds_{\xi}(R_{ik}) - \delta^h_k \pounds_{\xi}(R_{ij}) + \pounds_{\xi}(g_{ik} R^h_j) - \pounds_{\xi}(g_{ij} R^h_k)). \tag{72}$$

Since the spacetime admits Conh CC and Conh M, then, using Equations (8)–(10) and (40) in Equation (72), we obtain

$$\pounds_{\xi} Z^h_{ijk} = 0. \tag{73}$$

Now, applying the Conh CI Equation (20), we obtain

$$Z^h_{ijk} = 0 \qquad \text{(since } \alpha \neq 0\text{)}. \tag{74}$$

Thus, spacetime is conharmonically flat. $\square$

**Corollary 4.** *If a spacetime $(V_4, g)$ admits proper Conh CI and Conh CC, then the spacetime is conharmonically flat.*

**Proof.** The proof directly follows from Equation (63). $\square$

**Example 4.** *We consider a plane symmetric perfect fluid cosmological model obtained by Singh and Singh [24] that does not have a conformally flat spacetime. The geometry of this model is defined by the line element*

$$ds^2 = (1 + at)^2[-dt^2 + dx^2 + dy^2] + (1 + bt)dz^2$$

*where a and b are non-zero arbitrary constants. The above line element is found to admit a CIV, which is also a Conh CC $\xi^i = (A/a)\delta_0^i$ when a=b with $\alpha = -\frac{A}{(1+at)}$, where A is an arbitrary constant. However, when $a = b$, the model becomes conharmonic to flat spacetime and reduces to a special case $(k = 0)$ of the Friedmann–Robertson–Walker (FRW) model, representing a universe filled with disordered radiation.*

**Theorem 12.** *If a spacetime $(V_4, g)$ admits Conh M, then*

$$£_\xi Z_{ij} = 0. \tag{75}$$

**Proof.** Let a spacetime $(V_4, g)$ admit a conharmonic curvature tensor; then,

$$Z_{ij} = -\frac{1}{2}g_{ij}R. \tag{76}$$

Now, taking the Lie derivative of Equation (76),

$$£_\xi Z_{ij} = -\frac{1}{2}[(£_\xi g_{ij})R + g_{ij}(£_\xi(R)], \tag{77}$$

using Equation (67) with the condition of conharmonic motion, we obtain

$$£_\xi Z_{ij} = 0. \tag{78}$$

This implies that spacetime $(V_4, g)$ admits contracted conharmonic curvature collineation. □

## 5. Physical Interpretation to Fluid Spacetimes of General Relativity

In this section, we consider different types of fluid spacetimes as applications of Conh CI. If $(V_4, g)$ is a spacetime of the general theory of relativity with imperfect fluid (heat conducting and viscous) and a stress-energy tensor of the form:

$$T_{ij} = \mu u_i u_j + p h_{ij} - 2\sigma_{ij}\eta + u_i q_j + u_j q_i, \tag{79}$$

where projection tensor $h_{ij} = g_{ij} + u_i u_j$ and shear viscosity coefficient $\eta$ is non-negative, and the term $(2\sigma_{ij}\eta + u_i q_j + u_j q_i)$ in Equation (79) vanishes if $\sigma_{ij} = 0$ and $q^i = 0$ separately, then Equation (79) represents the stress-energy tensor for perfect fluid spacetime, i.e.,

$$T_{ij} = (\mu + p)u_i u_j + p g_{ij}. \tag{80}$$

In anisotropic fluid spacetime, the stress-energy tensor is of the form:

$$T_{ij} = \mu u_i u_j + p_\perp P_{ij} + p_\parallel n_i n_j, \tag{81}$$

where $p_\parallel$ and $p_\perp$ are the parallel and perpendicular components of the isotropic pressure to a unit vector $n^i$ orthogonal to $u^i$, respectively. $P_{ij} = h_{ij} - n_i n_j$ is the projection tensor onto the two orthogonal planes of vectors $u_i$ and $n_i$.

If $p = \frac{1}{3}(p_\parallel + 2p_\perp)$ and $2\sigma_{ij}\eta = (\frac{1}{3}h_{ij} - n_i n_j)(p_\parallel - p_\perp)$, then the form of the energy momentum tensor in anisotropic fluid is identical to imperfect fluid with $q^i = 0$.

Since self similar imperfect fluid spacetime admits homothetic vector $\xi^i$, i.e., self similarity is imposed on Equation (79), then the following equation holds [2]:

$$(a) \quad £_\xi \mu = -2\alpha\mu, \quad (b) \quad £_\xi p = -2\alpha p, \quad (c) \quad £_\xi u_i = -\alpha u_i,$$

$$(d) \quad £_\xi \sigma_{ij} = \alpha\sigma_{ij}, \quad (e) \quad £_\xi \eta = -\alpha\eta, \quad (f) \quad £_\xi q^i = -\alpha q^i. \tag{82}$$

From (82), we conclude that all physical quantities $(\mu, p, u_i, \sigma_{ij}, \eta, q_i)$ inherit the spacetime symmetry defined by $\xi^i$. Tupper and Coley [10] have investigated the conditions for an imperfect fluid to inherited symmetry (82) for a SCKV. Saridakis [25] et al. have solved the problem of symmetry inheritance for a spacelike proper CKV and other types of symmetry. Furthermore, Duggal [2] has also investigated the conditions for imperfect fluid, perfect fluid and anisotropic fluid to inherited symmetry (82) for a CIV, and Z. Ahsan [26] has investigated the necessary and sufficient conditions for perfect fluid spacetimes to admit Ricci inheritance symmetry.

We shall now consider spacetimes that admit a CKV $\xi_i$, i.e.,

$$£_\xi g_{ij} = 2\alpha g_{ij}, \tag{83}$$

where $\alpha(x^i)$ is the conformal function. As this is known for a CKV $\xi$ in fluid spacetime, then the following result holds [27]:

$$£_\xi u_i = -\alpha u_i + v_i, \tag{84}$$

where $v_i$ is the spacelike vector orthogonal to $u_i$, i.e., $u_i v^i{=}0$. Maartens [27] et al. have shown that $v_i \neq 0$ generally, and is given by

$$v_i = 2\xi^j \omega_{ij} + \beta \dot{u}_i - h_i^j \beta_{,j}, \tag{85}$$

where the vorticity tensor is denoted by $\omega_{ij}$ and $\beta = -u^i \xi_i$. Fluid flow lines are mapped onto fluid flow lines by the action of $\xi^i$ if $v_i = 0$. They are also said to be "frozen in" curves to the fluid.

For a CKV $\xi^i$ [10], the following results hold :

$$£_\xi R_{ij} = -\Box\alpha g_{ij} - 2\alpha_{;ij}, \tag{86}$$

$$£_\xi R = -2\alpha R - 6\Box\alpha, \tag{87}$$

$$£_\xi T_{ij} = 2(\Box\alpha g_{ij} - \alpha_{;ij}), \tag{88}$$

and the Einstein field equations are in the form

$$G_{ij} = R_{ij} - \frac{1}{2} R g_{ij} = T_{ij}. \tag{89}$$

In this section, we shall prove some results for the perfect fluid, imperfect fluid and anisotropic fluid on spacetime $(V_4, g)$ that admit the Conh CI vector $\xi^i$.

**Theorem 13.** *Let an imperfect fluid spacetime admit Conh CI symmetry along a vector field $\xi$, where fluid flow lines are mapped conformally by $\xi$. Then, the following equations hold:*

$$(a) \quad £_\xi \mu = 0 \quad (b) \quad £_\xi p = 0 \quad (c) \quad £_\xi u_i = -\alpha u_i, \tag{90}$$

$$(a) \quad £_\xi \sigma_{ij} = \alpha\sigma_{ij} \quad (b) \quad £_\xi \eta = \alpha\eta \quad (c) \quad £_\xi q_i = \alpha q_i. \tag{91}$$

**Proof.** The contraction of the Einstein field Equations (89) leads to

$$T = -R \qquad or \qquad R = -T, \tag{92}$$

Similarly, from Equation (79),

$$R = \mu - 3p. \tag{93}$$

Now, using the dynamic result for $T_{ij}$ of imperfect fluid by Equation (79), it leads to (cf., [2]), i.e.,

$$\pounds_\xi \mu = -2\Box\alpha - 2\triangle - 2\alpha\mu - 2\alpha_{;ij}u^i u^j, \tag{94}$$

where $\triangle = q^i v_i$. It is seen that the fluid flow lines are mapped conformally by $\xi^i$. This implies that $v_i = 0$. Hence, $\triangle = 0$ and Equation (94) reduces to

$$\pounds_\xi \mu = -2\Box\alpha - 2\alpha\mu - 2\alpha_{;ij}u^i u^j. \tag{95}$$

For imperfect fluid, when using (EFE) (89) with conditions $u_i u^i = -1$, $\sigma_{ij}u^i = 0$ and $q^i u_i = 0$, we obtain

$$R_{ij}u^i u^j = (\mu - \frac{R}{2}) = (\frac{3p + \mu}{2}). \tag{96}$$

If we set,

$$\alpha_{;ij} = \frac{\alpha}{2}[\frac{R}{3}g_{ij} - 2R_{ij}], \tag{97}$$

then, from [2], every CIV is also a CKV. Theorem 8 implies that spacetime admits Conh CI symmetry. If we multiply Equation (97) by $u^i u^j$, and using Equation (96) and $u^i u_i = -1$ ($u^i$ is timelike), then we obtain

$$\alpha_{;ij}u^i u^j = \alpha(\frac{R}{3} - \mu) = -\frac{\alpha}{3}(2\mu + 3p). \tag{98}$$

In view of (44)(b), Equations (95) and (98) yield $\pounds_\xi \mu = 0$; this implies that $\mu$ is constant under Lie differentiation. The proof of Equation (90)(b) follows from

$$\pounds_\xi p = -\frac{2}{3}\triangle + \frac{4}{3}\Box\,\alpha - 2\alpha p - \frac{2}{3}\alpha_{;ij}u^i u^j, \tag{99}$$

and $\triangle = 0$, (44)(b), (93) and (98).

Using $v_i = 0$ in Equation (84), we obtain (90)(c).

Moreover, from [2], it follows that,

$$v_i = 0 \qquad \Rightarrow \qquad \pounds_\xi \sigma_{ij} = \alpha\sigma_{ij}, \tag{100}$$

which proves Equation (91)(a). For imperfect fluid spacetime (with $T_{ij}$ of the form (79)), we have [2]

$$\pounds_\xi (\sigma_{ij}\,\eta) = (\alpha\eta + \pounds_\xi\eta)\sigma_{ij} = \alpha\frac{(2\mu + R)}{6}g_{ij} - \alpha R_{ij} + \alpha\frac{(4\mu - R)}{3}u_i u_j + \alpha(q_i u_j + q_j u_i). \tag{101}$$

Contracting Equation (101) with $\sigma^{ij}$ and using (79), $\sigma_{ij}u^i = 0$, $\sigma_{ij}g^{ij} = 0$ and Einstein field Equation (89), we obtain

$$(\pounds_\xi\eta + \alpha\eta)(2\sigma^2) = 4\alpha\eta\sigma^2, \qquad \text{where } \sigma_{ij}\sigma^{ij} = 2\sigma^2, \tag{102}$$

which leads to $\pounds_\xi\eta = \alpha\eta$ i.e., (91)(b) is proved. Finally, we prove (91)(c):

$$q_i(Q^{-1}\pounds_\xi Q) = -w_i \qquad \text{where } Q = q^i q_i \text{ and } w_i q^i = 0. \tag{103}$$

Since the $T_{ij}$ of imperfect fluid is represented by Equation (79), we have [10]

$$\pounds_\xi q_i = (Q^{-1}\pounds_\xi Q + \alpha)q_i + w_i, \tag{104}$$

from (103), Equation (104) leads to $£_\xi q_i = \alpha q_i$. $\square$

**Theorem 14.** *Let an imperfect fluid spacetime admit a Conh CIV $\xi^i$ with $(p + \mu) \neq 0$ and $q^i = 0$. Then,*

(a) *An eigenvector of $\alpha_{;ij}$ is $u^i$;*

(b) *$\xi^i$ is conformally mapped by fluid flow lines.*

**Proof.** For an imperfect fluid, using the Einstein field Equation (89) with conditions $u_i u^i = -1$, $\sigma_{ij} u^i = 0$ and $q^i u_i = 0$, we obtain

$$R_{ij} u^j = -(\mu - \frac{R}{2}) u_i = -(\frac{\mu + 3p}{2}) u_i. \tag{105}$$

Notice that, from Equation (105), $u^i$ is a timelike eigenvector of $R_{ij}$. After multiplying $u^i$ in Equation (97), and from (105) and (93), we obtain

$$\alpha_{;ij} u^j = (\frac{\alpha}{3})[3p + 2\mu] u_i, \tag{106}$$

which shows that $u^i$ is an eigenvector of $\alpha_{;ij}$; this proves the first part of the theorem. Now, using Equation (90)(c) in (84), we obtain $v_i = 0$, i.e, the vector $\xi$ is conformally mapped by fluid flow lines, and, hence, the proof of part (b) is complete. $\square$

**Theorem 15.** *Let a perfect fluid spacetime $(V_4, g)$ admit a Conh CIV $\xi$ with $(p + \mu) \neq 0$; then, the following equations hold:*

$$(a) \; £_\xi \mu = 0 \qquad (b) \; £_\xi p = 0. \tag{107}$$

**Proof.** First, contracting Equation (89), we obtain

$$T = -R \qquad or \qquad R = -T, \tag{108}$$

and then, contracting Equation (80), we obtain

$$R = \mu - 3p. \tag{109}$$

Next, we use a dynamic result for perfect fluid with $T_{ij}$ of the form (80) along a CKV vector field $\xi^i$ that was derived by Duggal in [2]:

$$£_\xi \mu = -2\Box\alpha - 2\alpha\mu - 2\alpha_{;ij} u^i u^j. \tag{110}$$

In a perfect fluid spacetime, using the (EFE) (89) with conditions $u_i u^i = -1$, $\sigma_{ij} u^i = 0$ and $q^i u_i = 0$, we obtain

$$R_{ij} u^i u^j = (\mu - \frac{R}{2}) = (\frac{3p + \mu}{2}). \tag{111}$$

If we multiply both sides by $u^i u^j$ in (97) and use Equation (111) and $u^i u_i = -1$ ($u^i$ is timelike), then we obtain

$$\alpha_{;ij} u^i u^j = \alpha(\frac{R}{3} - \mu) = -\frac{\alpha}{3}(2\mu + 3p). \tag{112}$$

Now using Equations (44)(b) and (112) in (110), we obtain $£_\xi \mu = 0$, i.e., Equation (107)(a) holds. Equation (107)(b) follows from

$$£_\xi p = \frac{4}{3}\Box\alpha - 2\alpha p - \frac{2}{3}\alpha_{;ij} u^i u^j. \tag{113}$$

Moreover, the use of Equations (44)(b), (112) and (109) in Equation (113) establishes the proof. □

**Theorem 16.** *Let a perfect fluid spacetime admit a Conh CIV $\xi^i$ and $(p + \mu) \neq 0$. Then,*

*(a)   An eigenvector of $\alpha_{;ij}$ is $u^i$;*
*(b)   Fluid flow lines are mapped conformally along the vector field $\xi^i$;*
*(c)   $\pounds_\xi u_i = -\alpha u_i$.*

**Proof.** The proof of the first part (a) is the same as the proof of the first part of Theorem 14. Now, we prove the second part of the theorem. By applying a dynamic result for a Conh CI vector in perfect fluid spacetime, we have [2]

$$(p + \mu)\, v_i = 2[(\alpha_{;kl}u^k u^l)\, u_i + \alpha_{;ik}\, u^k]. \tag{114}$$

Now, using Equations (106) and (112) in (114), we obtain

$$(p + \mu)v_i = 0 \quad \Rightarrow \quad v_i = 0 \qquad (\text{as,} \quad \mu + p \neq 0). \tag{115}$$

Finally, using Equation (115) in Equation (84), we obtain $\pounds_\xi u_i = -\alpha u_i$.

Now, we conclude that, by vector field $\xi$, the fluid flow lines are mapped conformally to Conh CI admitted by perfect fluid spacetime; consequently, the four-velocity vector $(u_i)$ is also inherited. □

**Theorem 17.** *Let anisotropic fluid spacetime $(V_4, g)$ admit a Conh CIV $\xi$ with $(P_\perp + \mu) \neq 0$ and $(P_\parallel + \mu) \neq 0$; then, the following equations hold:*

$$(a)\ \pounds_\xi \mu = 0, \qquad (b)\ \pounds_\xi P_\parallel = 0, \qquad (c)\ \pounds_\xi P_\perp = 0, \tag{116}$$

$$(d)\ \pounds_\xi u_i = -\alpha u_i, \qquad i.e.,\ \ v_i = 0. \tag{117}$$

**Proof.** For anisotropic fluid spacetime, the stress energy tensor is given by Equation (81). Now multiplying both sides of Equation (89) by $u^i u^j$ and $n^i n^j$, we obtain

$$R_{ij} u^i u^j = (\mu - \frac{R}{2}) \tag{118}$$

and

$$R_{ij} n^i n^j = (\frac{R}{2} + p_\parallel) \tag{119}$$

respectively. Moreover, from Equations (97) and (118), we have

$$\alpha_{;ij} u^i u^j = \alpha(\frac{R}{3} - \mu). \tag{120}$$

Since , for anisotropic fluid, $\mu$ must satisfy the following [2],

$$\pounds_\xi \mu = -2\square\alpha - 2\alpha\mu - 2\alpha_{;ij} u^i u^j. \tag{121}$$

From Theorem 5 (a), and Equation (120), Equation (121) reduces to (116)(a). The proof of the second part of (116) is followed by combining Equation (97) and (119); therefore,

$$\alpha_{;ij} n^i n^j = -\alpha(p_\parallel + \frac{R}{3}). \tag{122}$$

In anisotropic fluid, $p_\parallel$ must satisfy the following [2]:

$$\pounds_\xi p_\parallel = 2\square\alpha - 2\alpha p_\parallel - 2\alpha_{;ij} n^i n^j. \tag{123}$$

Again, using Equations (44)(a) and (122), Equation (123) reduces to (116)(b). The proof of the third part is as follows:

$$R_{ij}P^{ij} = 2p_\perp + R, \tag{124}$$

and using Equation (97), we obtain

$$\alpha_{;ij}P^{ij} = -2\alpha(p_\perp + \frac{R}{3}). \tag{125}$$

We also have [2]

$$\pounds_\xi p_\perp = 2\Box\alpha - 2\alpha p_\perp - 2\alpha_{;ij}P^{ij}. \tag{126}$$

If we put the value of $\Box\alpha$ and $\alpha_{;ij}P^{ij}$ in Equation (126), then (116)(c) holds, as we know that

$$\alpha_{;ij}p^{ik}u^j = 0, \qquad \alpha_{;ij}n^j u^i = 0. \tag{127}$$

For an anisotropic fluid, we have

$$(p_\parallel + \mu)v_j n^j = 2\alpha_{;ij}n^j u^i \tag{128}$$

and

$$(p_\perp + \mu)v_j p^{jk} = 2\alpha_{;ij}p^{jk}u^i. \tag{129}$$

Now, by virtue of Equations (127) and (128), Equation (129) reduces to

$$n^i vs_{.i} = 0 \quad \text{and} \quad p^{ij}v_j = 0, \qquad \text{where} \quad (\mu + p_\parallel) \neq 0, \quad (\mu + p_\perp) \neq 0. \tag{130}$$

We conclude that, from the above equations, $u^i$ and $v^j$ must be parallel. This result, combined with $v_i u^i = 0$, implies $v_i = 0$; thus, from (84), we have $\pounds_\xi u_i = -\alpha u_i$. $\quad\square$

**Theorem 18.** *A perfect fluid spacetime admits Conh CI along a conformal Killing vector field $\xi$ and also satisfies the EFE (1); then, the divergence of the conharmonic curvature tensor vanishes.*

**Proof.** Let $\xi$ be a Conh CI vector and also a CKV satisfying (20); then,

$$(R^{ij}\xi_j)_{;i} = -3\Box\alpha. \tag{131}$$

With the Einstein field Equations (90) and (44)(b), we obtain

$$((T^{ij} + \frac{R}{2}g^{ij})\xi_j)_{;i} = \alpha R. \tag{132}$$

Equation (132) explores a new equation of state for various matter. Perfect fluid spacetime satisfies (80) with $\xi \perp u$ or $\xi \parallel u$. Then,

$$((p + \frac{R}{2})\xi^i)_{;i} = \alpha R. \tag{133}$$

Now, we use $\xi^i_{;i} = 4\alpha$ and Equation (109) in the above equation to obtain $p + \mu = 0$; therefore, $Z^h_{ijk;h} = 0$ (cf., Theorem (2.1) in [4]). $\quad\square$

One can prove a similar result for an anisotropic fluid and imperfect fluid spacetime.

## 6. Conclusions

The idea of symmetry inheritance for a conharmonic curvature tensor is explored, and some related results are obtained on the Conh CI with both conformal motion and conharmonic motion in general and Einstein spacetime. We have obtained the necessary conditions for CI and conformal motion to have conharmonic curvature inheritance symmetry. We have also derived a result as a physical application for imperfect fluid, perfect

fluid and anisotropic fluid in the spacetime of general relativity. In the last result, it is concluded that the perfect fluid spacetime becomes either empty/Ricci flat, i.e., $(p + \mu = 0)$, or expresses the equation of state for a vacuum-like case, which is not a perfect fluid but is instead an Einstein spacetime.

**Author Contributions:** The authors contributed equally to this work. All authors have read and agreed to the published version of the manuscript.

**Funding:** Institute of Scientific Research and Revival of Islamic Heritage at Umm Al-Qura University, Saudi Arabia (Project # 43405050).

**Acknowledgments:** The authors are grateful to G. S. Hall, University of Aberdeen, Scotland for helpful discussions and suggestions. We are thankful to Md Danish Iqbal, Department of English, and S. S. Z. Ashraf, Department of Physics, AMU, for editing the language of the paper. The authors also wish to thanks the reviewers for constructive comments, which have led to extensive revision and improvement of the manuscript and acknowledge the finicial support by Institute of Scientific Research and Revival of Islamic Heritage at Umm Al-Qura University, Saudi Arabia.

**Conflicts of Interest:** The authors declare no conflict of interest. The funders had no role in the design of the study; in the collection, analyses or interpretation of data; in the writing of the manuscript, or in the decision to publish the results.

## Appendix A

### *Appendix A.1. Application to Cosmology*

Siddiqui and Ahsan [28] have studied the relativistic significance of conharmonically flat spacetime. A conharmonically flat spacetime is an Einstein spacetime that is consequently a space of constant curvature. The significance of the space of constant curvature is of great interest in the study of the cosmology (for further details, see [29]). For conharmonically flat spacetime, we have Equation (19):

$$Z^h_{ijk} = R^h_{ijk} + \frac{1}{2}(\delta^h_j R_{ik} - \delta^h_k R_{ij} + g_{ik}R^h_j - g_{ij}R^h_k) = 0. \tag{A1}$$

Contracting this, we obtain

$$R_{ij} = -\frac{1}{4}Rg_{ij}. \tag{A2}$$

Substituting this into the Einstein field Equation (1) with $\kappa = 1$, we obtain

$$3R_{ij} = T_{ij} \quad \text{or} \quad -\frac{3}{4}Rg_{ij} = T_{ij}. \tag{A3}$$

Many authors have found solutions to the modified field Equation (A3). However, there is a very important problem with these solutions.

We illustrate this by means of an example studied by Kumar and Srivastava [30]. For the FRW model,

$$ds^2 = -dt^2 + a(t)^2[\frac{dr^2}{(1 + kr^2)} + r^2(d\theta^2 + \sin^2\theta d\phi^2)], \tag{A4}$$

the field Equation (A3) yield

$$a\ddot{(t)} + [\frac{\mu}{9}]a(t) = 0, \tag{A5}$$

$$\frac{\ddot{a}}{a} + 2\frac{\dot{a}^2}{a^2} + 2\frac{k}{a^2} = \frac{p}{3}. \tag{A6}$$

In Equations (A5) and (A6), p and $\mu$ denote the pressure and density, respectively, of the perfect fluid (80), and k is an arbitrary constant. In addition, we see that Equation (A5) is satisfied for $a(t) = A\cos(\frac{\sqrt{\mu}}{3}t) + B\sin(\frac{\sqrt{\mu}}{3}t)$. Assuming $k = -2$, in the cases (i) A = 1 , B

= 0 (ii) A = 0, B = 1, Equations (A5) and (A6) have the common solutions when $p + \mu = 0$. This implies that the condition of the equation of state occurred for the FRW metric (A4).

If we further contract Equation (A2), we obtain the "vacuum", i.e., $R_{ij} = 0$. This is a very strong imposition. Thus, the additional symmetry requirement of conharmonic flatness reduces the space of solutions to "vacuum" solutions in general relativity.

*Appendix A.2. Conh CI with Conservation Law Generator*

Under the hypothesis of Theorem 9, spacetime $(V_4, g)$ possesses $R = 0$ and a Ricci tensor $R_{ij} \neq 0$ along a Conh Killing vector $\xi$ (Conh M) with the condition that $\xi$ satisfies Equation (68). Thus, it follows that

$$\pounds_{\xi} R = \pounds_{\xi}(R_{ij} g^{ij}) = (\pounds_{\xi} g^{ij}) R_{ij} = 0, \tag{A7}$$

where $\pounds_{\xi} g^{ij} = -g^{ik} g^{jl} \pounds_{\xi} g_{ij}$; then, Equation (A7) reduces to

$$R^{kl} \pounds_{\xi} g_{kl} = 0. \tag{A8}$$

Now, using $\pounds_{\xi} g_{kl} = \xi_{k;l} + \xi_{l;k}$ in (A8), we obtain

$$R_k^l \xi_{;l}^k = 0. \tag{A9}$$

From the twice-contracted Bianchi identity [8], we find (using $R = 0$)

$$R_{k;l}^l = 0. \tag{A10}$$

Combining Equations (A8) and (A10), we obtain

$$(R_k^l \xi^k)_{;l} = 0. \tag{A11}$$

In a spacetime with $R = 0$, the Einstein field Equation (1) take the form

$$R_k^l = \kappa T_k^l, \tag{A12}$$

where $\kappa$ is a constant and $T_k^l$ is an energy-momentum tensor with trace $T_l^l = T = 0$.

Substituting (A12) in (A11) gives

$$(\sqrt{g} T_k^l \xi^k)_{;l} = (\sqrt{g} T_k^l \xi^k)_{,l} = 0, \tag{A13}$$

where $g = |\det g_{ij}|$ and $\xi^k$ is defined by Conh CI. Thus, we conclude that, if a space-time $V_4$ with R = 0 and $R_{ij} \neq 0$ admits Conh CI along a Conh Killing vector $\xi$, then there exists a covariant conservation law generator of the form (A13).

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
