# Peer review of "Conharmonic Curvature Inheritance in Spacetime of General Relativityâ€"

_universe, doi:10.3390/universe7120505_

Round 1
Reviewer 1 Report
I appreciated the efforts of the authors made in the current version, but the manuscript is still written in a bad english style and it is still full of grammatical and syntax errors. The authors must completely edit the paper with the help of a native language expert in physics. Only after such a revision the paper can be accepted for publication.
Author Response
Kindly find the attached PDF file- Rebbutal merged with an edited manuscript herewith.

Reviewer 2 Report
The readability of the manuscript appears to be improved. Appendices have been added showing the application of the symmetry in question to solving problems of cosmology. Examples are given to demonstrate the presence of the conharmonic curvature inheritance symmetry in different cases. So the paper can be published now.
Author Response
Kindly find the PDF file- Rebbutal merged with the edited manuscript.

This manuscript is a resubmission of an earlier submission. The following is a list of the peer review reports and author responses from that submission.
Round 1
Reviewer 1 Report
The paper considers an interesting symmetry property, called conharmonic curvature inheritance symmetry in the spacetime, and analyzes the conditions for its occurrence in terms of the stress-energy tensor for certain kinds of matter. Mathematical constructions are described, and the results are proven.
As for the presentation, there are sentences with missing fragments or sentences that are grammatically incorrect or inconsistent with the notation in equations. For example, the phrase before eq (1.1),
"These are system of ten coupled highly nonlinear partial differential equations which is under consideration of without cosmological constant are given by:". May be, this should read
"It is a system of ten coupled highly nonlinear partial differential equations, which is under consideration and without the cosmological constant is given by:" or
"It is a system of ten coupled highly nonlinear partial differential equations, which is considered without the cosmological constant and is given by:"? After (1.1), there is the phrase
"...denote the of Riemannian metric tensor...". Is there something between "the" and "of" missing?
After (2.10), it is written
"where \alpha stands for the scalar function and holds given condition ...". But, first, there is no \alpha in eq (2.10); second, it might be better to write "and given condition ... holds"?
And so on.
It would also be useful to resume how occurrence of the considered symmetry simplifies solving the Einstein equations (additional conservation laws, a predefined form of the solution, etc.) and how typical this symmetry can be for real physical systems.
Author Response
The following are the corrections done according to the suggestions given by the reviewers.
PAGE 1
- Line no 10: Replaced “developed…… condition” by “prove that the conharmonic curvature tensor of a perfect fluid spacetime will be divergence-free”.
- Line No 17: Replaced “researcher” by “researchers”.
- Line No 23: Delete “Riemann”.
PAGE 3
- Line No 35: Replace “function” by “factor”
PAGE 4
- Line No 22: replace “alpha (α)” by “sigma (σ)”.
PAGE 10
- Line No 4: Replace “(4.7)” by “(4.8)”.
PAGE 23
- Line No 17: Replace “p=µ=0” by “p+µ=0”.
NOTE: In the whole manuscript the English language is complete reviewed and corrected. The revised version is submitted through the online revision process.
Reviewer 2 Report
See file in attachment

Author Response

(The authors gave the same response as above.)

Round 2
Reviewer 2 Report
In the revised version authors corrected only a very small part of the errors in the previous version. The revised version is full of errors and as a result its reading and understanding remains difficult for me. Also in the revised form I think that the paper is not suitable for publication.